# Can Clinical Assessment of Postural Control Explain Locomotive Body Function, Mobility, Self-Care and Participation in Children with Cerebral Palsy?

**DOI:** 10.3390/healthcare12010098

**Published:** 2024-01-01

**Authors:** Blanka Vlčkova, Jiří Halámka, Markus Müller, Jose Manuel Sanz-Mengibar, Marcela Šafářová

**Affiliations:** 1Department of Rehabilitation and Sports Medicine, 2nd Medical Faculty, Charles University and Motol University Hospital, 150 06 Prague, Czech Republic; blanka.vlckova@fnmotol.cz (B.V.); jiri.halamka@fnmotol.cz (J.H.); marcela.safarova@fnmotol.cz (M.Š.); 2Physiotherapy Department, Evangelisches Krankenhaus Düsseldorf Sozialpädiatrisches Zentrum, 40217 Düsseldorf, Germany; markusremueller@web.de; 3Queen Square Centre for Neuromuscular Diseases, University College London and National Hospital for Neurology and Neurosurgery, Queen Square, London WC1N 3BG, UK

**Keywords:** cerebral palsy, postural balance, locomotor activity, self-care, social participation

## Abstract

Trunk control may influence self-care, mobility, and participation, as well as how children living with cerebral palsy (CP) move around. Mobility and Gross Motor performance are described over environmental factors, while locomotion can be understood as the intrinsic ontogenetic automatic postural function of the central nervous system, and could be the underlying element explaining the relationship between these factors. Our goal is to study the correlation among Trunk Control Measurement Scale (TCMS) and Pediatric Evaluation of Disability Inventory (PEDI) domains, as well as Locomotor Stages (LS). Methods: A feasibility observational analysis was designed including 25 children with CP who were assessed with these scales. Results: The strong correlation confirms higher levels of trunk control in children with better self-care, mobility and participation capacities. Strong correlations indicate also that higher LS show better levels of PEDI and TCMS domains. Conclusions: Our results suggest that more mature LS require higher levels of trunk control, benefitting self-care, mobility and social functions.

## 1. Introduction

All the efforts of researchers and clinicians to capture postural control as the key link between distal manual function and propulsion capacity described in different forms (mobility, locomotion [1] or gross motor level), may be unclear due to the inability to extract the automatic motor outputs that are less dependent on the afferent inputs or intellectual intervention. Postural control can be defined as the control of the body position in space for the dual purposes of stability and orientation [2], but current neuroscience finds it difficult to capture and quantify in all of its dimensions [3]. Sometimes simplified as balance, considering only the ability to control the center of mass in relationship to the base of support, postural orientation includes the ability to maintain an appropriate relationship between the body segments, and between the body and the environment for a task [4].

Environmental performance in cerebral palsy (CP) depends on several factors; for this reason, most studies delimitate the assessment to standardized measures in clinical settings [5]. Understanding the performance in the daily environment of patients, from the capacities observed in a standardized environment [6,7], is a recurrent topic in research [8]. Measures of performance require long assessments and questionnaires, offering a wide picture but not an understanding of the essential factors affecting performance. Understanding these common elements affecting a different environmental performance would save time and will create a focus to target them therapeutically. Factors impacting the daily performance include mobility level, hand function and intellectual ability, and the current research focuses on how their interaction translates functionally in different environments. Higher levels of mobility [9], gross motor [10,11,12] and locomotor function [13] seems to relate to better self-care capacities and participation [14], but intellectual ability and age may play a role in this [15]. Different results among these studies found that gross motor ability was not strongly correlated to with self-care, as it is only significant in older children with CP from the age of seven years old, when social and intellectual aspects may play a role; on the other hand, the performance of self-care activities has not been found to require a high intellectual ability [9]. Incongruent results have also been previously described about whether the level of manual function is unrelated [9] or correlated [16,17,18] to the self-care performance. In addition, opposite opinions have been found about whether the handling performance is unrelated [19,20] or correlated [9] to the level of the gross motor function.

The Locomotor Stage (LS) is a reliable classification system that is sensitive and specific for gross motor function in typically developed children, as well as those living with CP [8]. LS was used for clinical evaluation and research to quantify therapeutic outcomes in patients with CP [21] because it combines assessing, prognostic and classificatory properties [8]. Within the framework of the International Classification of Functioning, Disability and Health (ICF) [2], LS takes the body postural function and independent activity components into account [4]. Research investigating spine postural control and gait capacity has also benefited by LS, because it allows plotting typically developed children with those living with CP [22]. These stages kinematically described the automatic human postural development, including hand grasping and upper limb weight bearing function. While Manual Ability Classification System (MACS) quantifies the self-generated hand function and the level of assistance or adaptation during daily performance, LS consider upper limb tasks in the context of a full-body movement directed to orientation or locomotion goals [8]. Upper-limb functioning must play a role in self-care; however, capturing this may depend on whether the manual ability is assessed in isolation (MACS) or longitudinally to the postural control framework [8]. The focus on the automatic ontogenetic dimension of postural control may also minimize the impact of cognition in their acquisition.

On the other hand, the Pediatric Evaluation of Disability Inventory (PEDI) assessment includes three domains within the framework of the International Classification of Functioning, Disability and Health (ICF) [6]. The self-care domain describes life tasks required for self-care within activities of daily living (ADL). The mobility domain focuses on environmental gross motor function. The cognition domain explores the participation components for a social performance [6]. PEDI showed a concurrent validation with other scales, quantifying gross and fine motor skills [23,24].

Our first goal is to understand how trunk control determines mobility, self-care and social capacities, by understanding the relationships amongst the Trunk Control Measurement Scale (TCMS) and PEDI scores. The second aim is to explore the criterion-related feature of the LS with gold standard scales for trunk control (TCMS), as well as self-care, mobility and social functions (PEDI).

## 2. Materials and Methods

A feasibility observational analysis was designed by a group of consultant physiotherapist specialists for assessing and treating children with cerebral palsy, after the content validity of the LS was reviewed by a panel of eleven experts. This panel focused on the reliability of the measurement by developing the clinical meaning of the qualitative and quantitative description of the Locomotor Stages. Some members also participated in previous research exploring its criterion reliability, sensitivity and specificity [8]. Patients were finally assessed by three pediatric physiotherapists with an average of 15 years of experience in this field, in two different rehabilitation centers.

### 2.1. Sample Size

The prevalence of CP is 2.08 cases/1000 births. To obtain a good security of the sample, 30 subjects for each of the groups to be evaluated would be required. As a feasibility study, our target was to reach about that number for the whole sample, but this was not reached for technical reasons. Preliminary calculations were carried out to understand the meaning of our measurements.

### 2.2. Participants

A total of 25 children met the inclusion criteria and no data were required to be excluded. Inclusion Criteria: children aged 4 to 18 years old with neuromotor findings consistent with CP [25], and Gross Motor Function Classification System (GMFCS) level I-V. Parents must agree to signing an informed consent from, as well as provide sufficient medical records. Exclusion criteria: neuromuscular disorders or other conditions that could affect motor performance, associated autistic spectrum disorders, or epilepsy, as well as visual and hearing impairments that could interfere with the testing.

Subjects were classified according to their GMFCS (Table 1), and were also observed according to the number of Bilateral and Unilateral CP subtypes according to the European committees responsible for this condition [26,27]. The average age of the sample was 9.6 ± 3.7 years old, distributed between the youngest participant, at 4 years of age and the oldest one, at 18 years of age.

### 2.3. Procedure

The clinical evaluation was conducted by two assessors at the same time at the Motol University Hospital (Czech Republic), to guarantee consistence and reliability, and by one assessor at the Sozialpädiatrisches Zentrum “Evangelisches Krankenhaus Düsseldorf” (Germany). The reliability and consistency between centers was achieved because one of the evaluators of each center was a member of the expert panel.

### 2.4. Outcomes

The following three outcome measures were quantified at specific times:

#### 2.4.1. Locomotor Stages According to Vojta

LS is a reliable classification system that is sensitive and specific for gross motor function [8]. LS is quantified by the highest locomotor pattern observed during a self-generated mobility without wearing any orthotics. The subject can be encouraged through play, but must not be assisted by another person or mobility aids.

#### 2.4.2. PEDI

PEDI is a reliable and validated assessment quantifying three ICF domains: self-care, mobility and social performance [22,23]. This questionnaire comprises a total of 197 items, which are assessed on an ordinal scale (0 = unable, 1 = able). This raw score is scaled within a range, between 0 = lowest function and 100 = highest function.

#### 2.4.3. TCMS

TCMS assessment tool shows a good relative reliability in children with CP. The highest TCMS score is 58 points, indicating a better trunk stability in sitting. The atatic sitting balance includes a maximum of 20 points. The selective control domain has a maximum of 28 points, and dynamic reaching has a maximum of 10 points.

### 2.5. Statistical Analysis

Data were analyzed in the “Paediatric Rehabilitation and Sports Medicine Department” at the Motol University Hospital (Czech Republic). Statistical analyses were performed using Github software package for Windows. Spearman’s rank correlation coefficient was calculated to understand the dependence among LS, PEDI and TCMS.

## 3. Results

LS scores in our sample can be observed in Table 1, showing that subjects were not equally distributed throughout all stages, and none of them classified as 0, 2 or 3. Table 2 shows the distribution of the negative correlation between LS and GMFCS (−0.81, *p* < 0.05). Higher levels in GMFCS and lower levels in LS stand for poor motor performance and seem to be equivalent.

Distribution of LS, TCMS and PEDI scores and subscores of each subject can be observed in Table 2, together with their classification according to their gross motor level GMFCS. For TMCS, the highest score was found in subject number 20, with 46 out of 58 points; the lowest score 0 was obtained by subjects 7 and 14.

Correlations: Spearman’s coefficient among the different domains of PEDI, LS and TCMS can be found in Table 3. Strong positive correlations observed in our study confirm higher levels of trunk control in children with better self-care, mobility and participation capacities. Strong positive correlations also indicate that higher LS show better levels of self-care and mobility, as measured by PEDI, as well as trunk stability, selective control and the dynamic range in sitting. The correlation between LS and social participation is moderate, and although significant, it is remarkably lower than the rest of the relationships among the studied variables. In addition, strong positive correlations have also been found among all the subscales of PEDI and TCMS.

## 4. Discussion

### 4.1. Ontogenetic and Automatic Postural Body Function

The way all aspects of health, functioning and disability are connected change from a linear to bi-directional perspective, acknowledging a dynamic interaction among entities such as “Body structures and function”, “activity” and “Participation” [6]. Therefore, interventions in one domain have the potential to modify one or more of the other entities, and this may be the reason why the clinical assessment of the ontogenetic postural body function could explain environmental self-care, mobility and participation, as well as postural control. Although some authors found that the gross motor capacity (standardized environment) was unrelated to everyday functioning [28], most of the research described how the gross motor capacity is considered an important factor for mobility in daily life [12,29,30]. The capacity, capability and performance [7] “in locomotor terms” must have some intrinsic elements in common, which we named as the “ontogenetic automatic postural body function” (Figure 1). Because personal and external factors vary and are difficult to control, this underlying “genotype” body function might not always be expressed typically (Locomotor Stages); however, the postural essence is expressed typically. Positive results observed in interventional studies targeting the ontogenetic development of postural control, without utilizing functional training, seem to support this current ICF view [21].

### 4.2. Locomotor Ontogenic Stages vs. Motor Milestones

On the other hand, there is some discussion in developmental guidelines about the role of traditional motor milestones. For instance, crawling has recently been removed from some surveillance checklists, as its observation is inconsistent across different cultures, with a lack of normative kinematics and variability of onset. Therefore, it was concluded that crawling is “not essential for development” [31]. This statement is based on the kinematic description of this locomotion, while our view is to focus on a neurodevelopmental “stage”: the onset of a more mature inner-level of the postural control of the Central Nervous System. Crawling is not essential for development since its practice does not improve the quality of further milestones [31]. On the contrary, the way LS approaches neurodevelopment is by understanding that older children who have achieved more mature levels of locomotion like walking, will never be able to replicate the crawling that they showed when walking was not available. The ontogenetic LS 5 “Reciprocal Crawling” cannot be practiced by an older child or an adult when moving around on their hands and knees, because this was an automated neural coordination that required no previous training, but just the adequate level of motivation to move forward at a particular maturation stage. Shortly after the LS 5 onset, practice and age will offer more variability and adaptation to an output that was initially triggered similarly to a very restricted pattern of movement. Children with CP whose highest LS is 4 or 5 (Pathologic or Reciprocal crawling) also do not use the Ontogenetic LS when moving around on hands and knees; however, they use an atypically adapted pattern, with further levels of maturation and volition. If we consider CP as a “blocking factor” for the ontogenetic developmental output, we can also identify other minor conditions responsible for the crawling to be “inconsistently observed”. This may include environmental deprivation (cultural or social) or intrinsic factors (sensory impairments; level of motivation—cognition; or musculoskeletal factors, generating the variability of crawling kinematics). This lack of “observable expression” of a developmental stage does not imply that the child has not developed that stage neurologically, and this supports the conclusion of why “crawling is not essential”, but we would complete it with the phrase: “to be observed or trained.” Many physiotherapists around the world are divided by this controversial topic, and this understanding may unify interdisciplinary teams, because we all agree on providing the richest conditions as possible, regardless of whether the expression of crawling or other milestones is observed clinically. General gross motor milestones are met across different cultures according to the World Health Organization, who studied 2–24-month-old children from five different backgrounds and described how 90% of the infants achieved 5/6 gross motor milestones [32]. This exposed the genetic factor guiding partially gross motor development (Figure 1), which is capable of adapting to individual features and environmental demands. Although both components are clearly relevant, the current research seems to have clarified how to target external factors therapeutically (motor learning and goal-directed activities), while other interventions try to understand how to tackle the genetically preprogrammed components [21].

### 4.3. Manual Function in Self-Care and Locomotion

Although mobility and participation has been described as significant factors influencing self-care, mobility alone could explain self-care activities in children older than 7 years of age [9]. The reason why locomotion could reflect self-care aspects that, at first glance, seem to have more to do with manual ability or age than with mobility, could be in the ontogenetic foundations of LS. During the first year of human development, the manual function seems to be strongly related to postural development [8,33]. LS quantifies human ontogenetic postural development by describing increasing levels of postural control from the kinematics of different locomotor capacities, from birth to independent walking. This includes grasping and other upper limb capacities (like weight bearing on the elbow or hand) in a linear way (instead of in parallel, like in Manual Ability Classification System), which arise along with more mature levels of postural control [8,34] (Appendix A).

### 4.4. Clinical Implications and Limitations

Our strong negative correlation confirmed the equivalency of reverse growing between GMFCS and LS, which were observed in previous research [8]. Lower levels in GMFCS and higher levels in LS stand for better and equivalent motor performance. Besides the limited sample, the correlation tendency between LS and PEDI may be funded in the categorization of ten locomotor stages. Ten clinically meaningful changes could be detected with this scale, which were equivalent to the PEDI Minimally Clinically Important Difference (MCID) of the 11% average variation across all of subscales [35]. Future regression calculations including larger samples could help to clarify this distribution.

Our current study also glimpses how postural control body function (complex to measure in all dimensions) can be reflected in the locomotor capacity of children with CP (which is easily quantified in 10 stages). The TCMS test took approximately 20 min to administer and PEDI test about 45–60 min, as it is time consuming for therapists, while LS can be obtained during the usual clinical observation required for diagnostic or therapeutic reasons. The clinical implications of understand the relationships between LS scores and both previous scales would be for therapists, physicians, and researchers to make a sensitive prognosis over daily self-care, mobility and social performance from single motor patterns that are easily assessed in the therapy room.

The limitation of our feasibility study is the reduced number of subjects assessed. The focus on the automatic ontogenetic dimension of the postural control of the LS, which intrinsically includes a developmental age, may also minimize the impact of cognition and social aspects in the acquisition of self-care, mobility and even social participation. The moderate significant correlation between the LS and PEDI participation subscale suggests this, but stronger values among the other variables cast a shadow over this result. To capture this, and complete the distributions suggested in this work, larger samples are recommended that include children in every stage of development and age range.

## 5. Conclusions

The strong positive correlation observed in our study points to higher levels of trunk control in children with better self-care, mobility and participation capacities. In addition, strong positive correlations obtained when exploring the criterion-related validity of the LS suggest that more mature stages of locomotor capacities require higher levels of trunk control, benefitting self-care, mobility and social functions.

## Figures and Tables

**Figure 1 healthcare-12-00098-f001:**
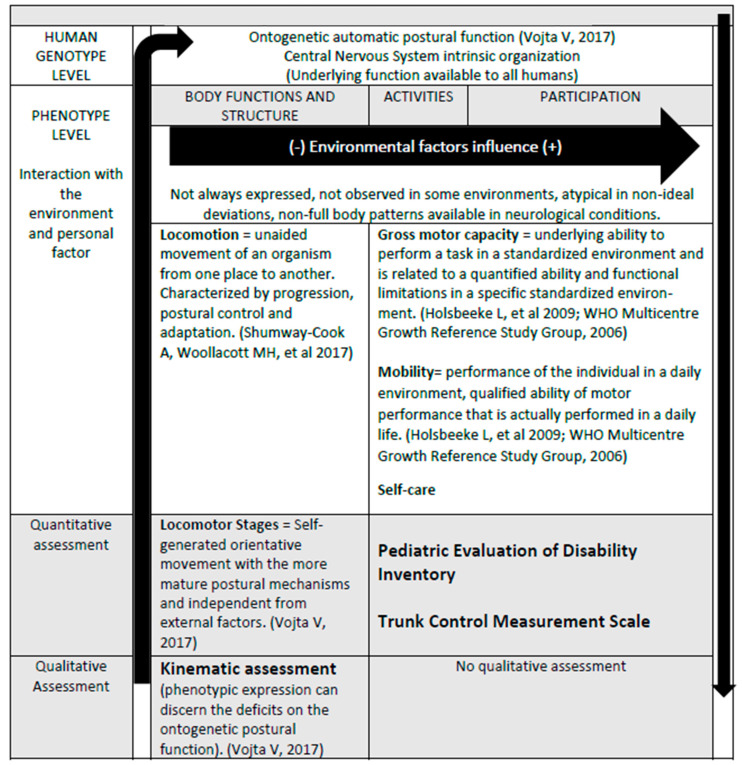
Ontogenetic Postural Function and Locomotor Stages. Shumway-Cook A, Woollacott MH, et al 2017; Holsbeeke L, et al 2009; WHO Multicentre Growth Reference Study Group, 2006; Vojta V, 2017.

**Table 1 healthcare-12-00098-t001:** Sample distribution and classification.

n = 25		Level	Quantity	%
**Gross Motor**		I	7	28
**Function**		II	6	24
**Classification**		III	6	24
**System**		IV	3	12
**(GMFCS)**		V	3	12
**Locomotor**		0	0	0
**Stages**		1	3	12
		2	0	0
		3	0	0
		4	2	8
		5	1	4
		6	4	16
		7	5	20
		8	7	28
		9	3	12
**Cerebral**	**Bilateral**	Quadriparesis	5	20
**Palsy**		Diparesis	11	44
**Subtypes**		Triparesis	1	4
	**Unilateral**	Hemiparesis	8	32

**Table 2 healthcare-12-00098-t002:** Sample classification according to their Gross Motor Function Classification System (GMFCS). Description of each subject according to their Locomotion Stages (LS), Trunk Control Measurement Scale (TCMS) and Pediatric Evaluation of Disability Inventory (PEDI).

			TCMS				PEDI		
Subject Number	GMFCS	LS	Static Sitting Balance TCMS (20)	Selective Control TCSM (28)	Dynamic Range TCMS (10)	TCSM Overall Score (58)	Scaled Score Self-Sufficiency	Scaled Score Mobility	Scaled Score Social Features
1.	III	6	11	6	3	20	60.5	52.2	64.1
2.	IV	4	8	4	3	15	59.9	30.6	82.2
3.	III	7	13	11	7	31	100	85.2	82.2
4.	IV	5	6	3	3	12	57.4	37.1	66.2
5.	I	6	10	9	6	25	62.5	79.8	82.2
6.	III	6	13	17	9	39	85.1	63.9	89.1
7.	V	1	0	0	0	0	35.1	0.0	53.2
8.	V	1	3	1	4	8	43.6	49.7	38.8
9.	II	7	18	9	8	35	60.5	79.8	77.3
10.	I	7	13	11	6	30	70.8	94.2	67.4
11.	II	7	18	12	5	35	100.0	89.2	96.3
12.	III	8	16	9	3	28	59.3	54.8	73.4
13.	I	8	13	12	3	28	69.1	77.3	65.1
14.	V	1	0	0	0	0	30.7	6.1	35.1
15.	II	8	20	11	7	38	68.3	94.2	82.2
16.	II	8	20	10	6	36	63.2	61.9	82.2
17.	IV	4	3	2	0	5	43.6	46.1	49.7
18.	III	7	20	15	9	44	63.2	61.9	82.2
19.	II	8	20	13	6	39	54.3	68.7	59.9
20.	I	9	20	18	8	46	100	100	100
21.	III	6	14	6	0	20	65.2	77.3	89.1
22.	I	9	20	15	9	44	100	100	100
23.	I	9	20	16	8	44	100	100	100
24.	I	8	20	16	10	46	100	100	100
25.	II	8	20	14	9	43	81.4	65.0	68.9

The maximum number of points that can be obtained in the specific part of TCSM is presented in parentheses.

**Table 3 healthcare-12-00098-t003:** Correlation among different domains of PEDI, LS and TCMS.

		LS	TCMS Overall Score	PEDI Self-Care	Mobility	Participation
**PEDI**						
	Self-Care	0.68 *	0.76 *		0.84 *	
	Mobility	0.75 *	0.75 *			0.72 *
	Participation	0.58 *	0.72 *	0.83 *		
**TCMS**		0.84 *				
	Static sitting balance	0.89 *		0.64 *	0.69 *	0.64 *
	Selective control	0.79 *		0.82 *	0.74 *	0.70 *
	Dynamic range	0.63 *		0.69 *	0.66 *	0.62 *

* *p* < 0.05. PEDI = Pediatric Evaluation of Disability Inventory. TCMS = Trunk Control Measurement Scale. LS = Locomotor Stages.

## Data Availability

The supporting data reported can be provided upon request to the corresponding author.

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
