# Peer review of "Can Clinical Assessment of Postural Control Explain Locomotive Body Function, Mobility, Self-Care and Participation in Children with Cerebral Palsy?"

_healthcare, 2024, doi:10.3390/healthcare12010098_

Round 1
Reviewer 1 Report (Previous Reviewer 1)
Comments and Suggestions for Authors
Thank you for inviting me again to be a reviewer of your journal. I reviewed the manuscript as much as I could.
In this manuscript, the authors reported on the relationships between Locomotor Stages (LS), Trunk Control Measurement Scale (TCMS), and Pediatric Evaluation of Disability Inventory (PEDI) in children with cerebral palsy. The authors conclude that "The clinical implications of understand the relationships between LS scores and both previous scales (TCMS and PEDI) would be for therapists, physicians, and researchers to make sensitive prognosis over daily self-care, mobility and social performance from single motor patterns easily assessed in the therapy room."
I would like to pay tribute to the authors for their efforts in assessments of children with cerebral palsy.
I think the manuscript has been appropriately revised and contains important data and suggestions. I consider this paper to be ready for acceptance for publication in your journal.
Author Response
Dear Editor and reviewers of the manuscript entitled “Can clinical assessment of postural control explain locomotive body function, mobility, self-care and participation in children with cerebral palsy?”.
REVIEWER 1
Thank you for inviting me again to be a reviewer of your journal. I reviewed the manuscript as much as I could.
In this manuscript, the authors reported on the relationships between Locomotor Stages (LS), Trunk Control Measurement Scale (TCMS), and Pediatric Evaluation of Disability Inventory (PEDI) in children with cerebral palsy. The authors conclude that "The clinical implications of understand the relationships between LS scores and both previous scales (TCMS and PEDI) would be for therapists, physicians, and researchers to make sensitive prognosis over daily self-care, mobility and social performance from single motor patterns easily assessed in the therapy room."
I would like to pay tribute to the authors for their efforts in assessments of children with cerebral palsy.
I think the manuscript has been appropriately revised and contains important data and suggestions. I consider this paper to be ready for acceptance for publication in your journal.
We would like to thank you for your comments and for allowing us to address the issues you raised to improve the manuscript’s quality. We appreciate your observations and the time devoted to the constructive criticism and feedback of our manuscript.
Reviewer 2 Report (New Reviewer)
Comments and Suggestions for Authors
I appreciate the significance of your study in addressing the relationship between trunk control and various functional domains in children with cerebral palsy. Your work has the potential to contribute valuable insights to the field of pediatric rehabilitation . I have completed a thorough review of the manuscript and have provided detailed comments and suggestions for improvement. Please find my feedback below:
1- The rationale behind selecting the feasibility approach is not clear enough .almost we select the visibility study to preliminary explorae the proposed project and the selcted outcomes for this type of study such as such as legality ,saftey,should serve this purpose
2- Sample Size Limitations:The study acknowledges that the target sample size was not achieved due to technical reasons. Without specification of these reasons. The limitations of the sample size should be more discussed , and the impact on the internal and external validity should be addressed.
3- Inclusion and Exclusion Criteria:The selected age is very broad ,more details about the inclusion and exclusion certeria should be added
4- Observer Bias and Reliability: The study involves multiple assessors, please clarify how the reliability of the assessments and the consistency of the results were achiveed ?
5- Incomplete Reporting on LS: more details about the Locomotor Stages (LS) as a classification system, should be added
6- Limited Discussion on Cognitive Aspects:many confounding factors should acknowledged such as the cognitive aspects, age, severity of CP, and comorbidities ,….which has significant impact on all outcomes i, a more details regarding these these factors are warranted.
7-Underdeveloped Implications for Clinical Practice:
clinical impact of research is overlooked ,more discussion about the clinical impact of your research should be addresed
Author Response
Dear Editor and reviewers of the manuscript entitled “Can clinical assessment of postural control explain locomotive body function, mobility, self-care and participation in children with cerebral palsy?”.
First of all, we would like to thank you for your comments and for allowing us to address the issues you raised to improve the manuscript’s quality. We appreciate your observations and the time devoted to the constructive criticism and feedback of our manuscript. Please find the answer to your comments below and the recommended changes have been highlighted in blue in the manuscript.
REVIEWER 2
- The rationale behind selecting the feasibility approach is not clear enough .almost we select the visibility study to preliminary explorae the proposed project and the selcted outcomes for this type of study such as such as legality ,saftey,should serve this purpose
We agree that feasibility studies aim to understand safety and security aspects. Although this is a limitation of our study, we found important to show our arising correlations even with a small sample and such a disparity of data / scales. Even if we cannot extrapolate our conclusion, the goal of this initial humble study was not to understand the regression or make predictions, but introduce the concept of “Ontogenetic automatic postural body function” as a common foundation for these scales. We also had a budget / capacity limitation for this initial work (please see our comments on your next point about sample size), that could be solved with increasing interest in this topic. In relation to this, we could support the limitation of our study with the following statements.
Although pilot studies will not have sufficient sample sizes to test measurement equivalence, investigators can review literature describing performance in diverse groups. Identifying measures with evidence of conceptual and psychometric adequacy in the target population increases the likelihood that only minimal feasibility testing will be necessary. Feasibility testing can focus on multiple primary outcome measures to determine if one or more are not acceptable or understood as intended. including sample sizes for pilot studies, estimates of minimally important differences, design effects, confidence intervals and non-parametric statistics. An in-depth treatment of the limits of effect size estimation as well as process variables is presented. Teresi JA, Yu X, Stewart AL, Hays RD. Guidelines for Designing and Evaluating Feasibility Pilot Studies. Med Care. 2022 Jan 1;60(1):95-103. doi: 10.1097/MLR.0000000000001664. PMID: 34812790; PMCID: PMC8849521
Sample size should be based on “practical considerations including participant flow, budgetary constraints, and the number of participants needed to reasonably evaluate feasibility goals.” For qualitative work, to reach saturation, sample sizes may be 30 or less. For quantitative studies, a sample of 30 per group (intervention and control) may be adequate to establish feasibility (29). Whitehead AL, Julious SA, Cooper CL, et al. Estimating the sample size for a pilot randomised trial to minimise the overall trial sample size for the external pilot and main trial for a continuous outcome variable. Stat Methods Med Res 2016;25(3):1057–1073.
- Sample Size Limitations:The study acknowledges that the target sample size was not achieved due to technical reasons. Without specification of these reasons. The limitations of the sample size should be more discussed , and the impact on the internal and external validity should be addressed.
Lack of resources / capacity was the reasons for stopping recruitment. A granted interventional trial took over and limit our capacity to keep measuring children in both centres. The word “technical reasons” may bring confusion, and “grant” may also be unclear, so we have now stated “due to resources capacity”.
Sample size should be based on “practical considerations including participant flow, budgetary constraints, and the number of participants needed to reasonably evaluate feasibility goals.” For qualitative work, to reach saturation, sample sizes may be 30 or less. For quantitative studies, a sample of 30 per group (intervention and control) may be adequate to establish feasibility (29). Whitehead AL, Julious SA, Cooper CL, et al. Estimating the sample size for a pilot randomised trial to minimise the overall trial sample size for the external pilot and main trial for a continuous outcome variable. Stat Methods Med Res 2016;25(3):1057–1073.
- Inclusion and Exclusion Criteria:The selected age is very broad ,more details about the inclusion and exclusion certeria should be added
The reasoning behind the age range was to reach all the GMFCS age ranges (0-2, 2-4, 4-6, 6-12 and 12-18). The lower limit was given by the reliability of TCMS from 4-5 years old.
In our limitations we have stated that “To capture this, and complete the distributions suggested in this work larger samples are recommended, including children in every stage of development and age range.”
- Observer Bias and Reliability: The study involves multiple assessors, please clarify how the reliability of the assessments and the consistency of the results were achiveed ?
The reliability and the consistency between centres were achieved because one of the evaluators of each centre belong to the members of the expert panel described on the methodology. Please find in the first Material and Methods Paragraph: “This panel focused on the reliability of the measurement, by developing the clinical meaning of the qualitative and quantitative description of the Locomotor Stages. Some members participated also in previous research exploring its criterion reliability, sensitivity and specificity [4]. Patients were finally assessed by three paediatric physiotherapists with an average of 15 years of experience in this field, in two different rehabilitation centres.”
In the centre with two evaluators, the assessments were performed together, to improve even more these parameters. Another reviewer has requested to hide the personal information about the evaluator as a common topic in scientific research, so the final paragraph has been adapted as followed:
“The clinical evaluation was conducted by two assessors at the same time at the Motol University Hospital (Czech Republic), to guarantee consistence and reliability, and by one at the Sozialpädiatrisches Zentrum “Evangelisches Krankenhaus Düsseldorf” (Germany). The reliability and the consistency between centers were achieved because one of the evaluators of each center belong to the members of the expert panel.”
5- Incomplete Reporting on LS: more details about the Locomotor Stages (LS) as a classification system, should be added
This has now been completed in the introduction section. Both introduction and discussion sections have now been reorganized to understanding of this.
6-Underdeveloped Implications for Clinical Practice:
clinical impact of research is overlooked ,more discussion about the clinical impact of your research should be addressed
Discussion section have now been reorganized and sub-sections have been added to improve fluidity and understanding. Clinical implications of our research have been now clearer stated.

Reviewer 3 Report (New Reviewer)
Comments and Suggestions for Authors
Thank you for allowing me to review the article titled “Can clinical assessment of postural control explain locomotive body function, mobility, self-care and participation in children with cerebral palsy?”. The purpose of this article was to examine the relationship between trunk control and performance in self-activity, mobility and participation in children with cerebral palsy, concluding that greater trunk control is associated with improvements in these areas, especially at later locomotor stages.
Introduction
- Ensure that ideas connect more fluidly, especially when discussing discrepancies in previous studies. Including clearer transitions may benefit.
- line 65 ends with a broken syllable. Please correct and revise the document.
Material and methods:
- There is a discrepancy in the numbering of subsections. For example, it goes from 2.1 to 2.1.1.1.2.2 without clear organization. It would be advisable to correct this for clarity and consistency.
- on page 2, line 94-99 is in italics. Please correct.
- Subjects were classified according to their GMFCS in table 1… “ in line 104, it would be clearer if it is put like this: “Subjects were classified according to their GMFCS (Table 1)…
- The explanation of the sample size is somewhat brief. It would be advisable to expand on the technical reasons that prevented the desired sample size from being reached and how this might affect the findings or conclusions of the study.
- In scientific articles, it is often preferable to focus on describing the places or institutions where the research is conducted rather than mentioning specific names of individuals, unless it is essential for the context or methodology. This helps to maintain a more neutral and professional approach to the presentation of information Lines 112-113).
Results
- the results section has only one section 3.1 . Perhaps it would be better to remove this numbering.
- The results part is too short. I suggest expanding it to provide a more detailed description of the results to improve the understanding and impact of the findings.
Discussion
-In the discussion part there are references in bold. please check.
- It would be useful to clarify how the strong negative correlation between GMFCS and LS confirms the equivalence of inverse growth, supporting previous findings. The significance of this pattern of inverse correlation and its relationship to the existing literature should be explained.
- In my humble opinion, the discussion is a bit difficult to read. Perhaps it would benefit from several sub-sections to make it clearer. We are talking about postural control and then, after a point and a continuation, we talk about limitations.
The section on limitations is a bit terse.
- It would be necessary to specify which ethics committee has been passed. What happened in Murcia?
References
- Reference 19 is in a different colour. Please correct to have the same format.
Author Response
Dear Editor and reviewers of the manuscript entitled “Can clinical assessment of postural control explain locomotive body function, mobility, self-care and participation in children with cerebral palsy?”.
First of all, we would like to thank you for your comments and for allowing us to address the issues you raised to improve the manuscript’s quality. We appreciate your observations and the time devoted to the constructive criticism and feedback of our manuscript. Please find the answer to your comments below and the recommended changes have been highlighted in blue in the manuscript.
Introduction
- Ensure that ideas connect more fluidly, especially when discussing discrepancies in previous studies. Including clearer transitions may benefit.
. Both introduction and discussion sections have now been reorganized to improve fluidity and understanding
- line 65 ends with a broken syllable. Please correct and revise the document.
We tried to change this, but it is not possible because this formatting is due to the use of the requested journal template.
Material and methods:
- There is a discrepancy in the numbering of subsections. For example, it goes from 2.1 to 2.1.1.1.2.2 without clear organization. It would be advisable to correct this for clarity and consistency.
We have changed this, which seems to be altered during the template uploading.
- on page 2, line 94-99 is in italics. Please correct.
We have now changed this, thank you for spotting this.
- Subjects were classified according to their GMFCS in table 1… “ in line 104, it would be clearer if it is put like this: “Subjects were classified according to their GMFCS (Table 1)…
We have now changed this.
- The explanation of the sample size is somewhat brief. It would be advisable to expand on the technical reasons that prevented the desired sample size from being reached and how this might affect the findings or conclusions of the study.
Lack of resources / capacity was the reasons for stopping recruitment. A granted interventional trial took over and limit our capacity to keep measuring children in both centres. The word “technical reasons” may bring confusion, and “grant” may also be unclear, so we have now stated “due to resources capacity”.
- In scientific articles, it is often preferable to focus on describing the places or institutions where the research is conducted rather than mentioning specific names of individuals, unless it is essential for the context or methodology. This helps to maintain a more neutral and professional approach to the presentation of information Lines 112-113).
These specifications were requested by a previous reviewer, but we agree on your neutral approach, and have removed them again. To satisfy both requests, we have now improved this paragraph making specification on how the reliability and consistency was achieved:
“The clinical evaluation was conducted by two assessors at the same time at the Motol University Hospital (Czech Republic), to guarantee consistence and reliability, and by one at the Sozialpädiatrisches Zentrum “Evangelisches Krankenhaus Düsseldorf” (Germany). The reliability and the consistency between centers were achieved because one of the evaluators of each center belong to the members of the expert panel.”
Results
- the results section has only one section 3.1 . Perhaps it would be better to remove this numbering. This formatting mistake has not been removed.
- The results part is too short. I suggest expanding it to provide a more detailed description of the results to improve the understanding and impact of the findings.
The results section was asked to be more summarized by another review, and therefore we tried to show the data within the tables and with less text. Nevertheless, we have now improve the description of the impact of the findings as requested.
Discussion
-In the discussion part there are references in bold. please check.
The journal request that the name of the journal should be in italics and the year of publication in Bold.
- It would be useful to clarify how the strong negative correlation between GMFCS and LS confirms the equivalence of inverse growth, supporting previous findings. The significance of this pattern of inverse correlation and its relationship to the existing literature should be explained.
This has now been clarified in the paragraph “Our strong negative correlation confirmed the equivalency of reverse growing between GMFCS and LS observed in previous research [4]. Lower levels in GMFCS and higher levels in LS stand for better and equivalent motor performance.”
- In my humble opinion, the discussion is a bit difficult to read. Perhaps it would benefit from several sub-sections to make it clearer. We are talking about postural control and then, after a point and a continuation, we talk about limitations.
Discussion section have now been reorganized and sub-sections have been added to improve fluidity and understanding.
The section on limitations is a bit terse.
- It would be necessary to specify which ethics committee has been passed. What happened in Murcia?
The reason behind Murcia ethics committee is that this was my PhD University base and where most of my research topics have develop. I still belong to this research group an collaborate in much research from there.
References
- Reference 19 is in a different colour. Please correct to have the same format.
Comments from previous reviewers were highlighted that way, and this has now been removed.

This manuscript is a resubmission of an earlier submission. The following is a list of the peer review reports and author responses from that submission.
Round 1
Reviewer 1 Report
Comments and Suggestions for Authors
General Comments:
In this manuscript, the authors reported on the relationships between Locomotor Stages (LS), Trunk Control Measurement Scale (TCMS), and Pediatric Evaluation of Disability Inventory (PEDI) in children with cerebral palsy. The authors conclude that "The clinical implications of understand the relationships between LS scores and both previous scales (TCMS and PEDI) would be for therapists, physicians, and researchers to make sensitive prognosis over daily self-care, mobility and social performance from single motor patterns easily assessed in the therapy room."
I would like to pay tribute to the authors for their efforts in assessments of children with cerebral palsy.
The study design itself is generally acceptable. It can be assumed that the results of the three evaluation methods correlate. However, this is the case when conditions other than trunk control are constant or not taken into account. The Body Function and Structures (which may include single motor patterns, or trunk control) is a different hierarchy than the Activities and Participation in ICF (which may include daily self-care, mobility and social performance). While the overall PEDI score may indeed correlate with trunk control, the manuscript does not mention what the individual daily activities of daily living are. Even without LS, the GMFCS may be sufficient for simple prediction of activity and participation. However, it is questionable whether the prediction using LS or GMFCS can be called true sensitive prediction.
Specific recommendations for revision
a) major points:
1. Participants (Line 97)
There are mixed levels of GMFCS, mixed subtypes of paralysis, and a large age range.
In addition to improving body functions and structure itself, rehabilitation interventions are aimed at improving activity and participation even in the presence of impairments. Wide age range may affect the results.
2. Statistical analysis (Line 125)
Shouldn't the correlation analysis for ordinal scales be done by Spearman instead of Pearson?
b) major points:
1. Line 19
Description?
2. Table 1 (Line 108)
Isn't the column “total” unnecessary?
3. Figure 1 (Line 241)
BODY FUNCTIONS AND STRUCTURE
Please confirm.
Reviewer 2 Report
Comments and Suggestions for Authors
First of all, we would like to thank the authors for their novel research in such an important and interesting area of knowledge as cerebral palsy. Before the manuscript can be considered for publication, the authors must address the following issues:
1. Bibliography: authors should adapt the bibliography to the requested journal format. Not in superscripts but between [] Example: [1]. In addition, in the references themselves the name of the journal should appear in italics and the year of publication in bold.
2. Abstract: Delete the word background.
Keyword: eliminate the word keyword 1 and the numbers: 1. 2. 3. 4. Make sure that the keywords are MESH terms.
4. Introduction: No comments.
5. Material and methods: This section should be restructured. I advise the authors to add the following subsections:
o 2.1. Sample size (To be completed).
o 2.2. Participants (Done)
o 2.3. Procedure (Indicate the evaluation procedure for each of the scales, where it was carried out, how and by whom, etc.).
o 2.4. Outcomes (add the psychometric properties of each scale used: PEDI and TCMS).
o 2.5. Statistical analyses: Clearly indicate the analysis performed so that readers have a clear understanding. Also indicate which category and terminology was used to express the degree of correlation (Salter, K., Jutai, J. W., Teasell, R., Foley, N. C., Bitensky, J., & Bayley, M. (2005). Issues for selection of outcome measures in stroke rehabilitation: ICF activity. Disability and rehabilitation, 27(6), 315-340).
6. Table 1: Table 1 is not referring to the sociodemographic characteristics of the sample. Sociodemographic characteristics would refer to chronological age, sex, height, weight, etc. etc. It is advisable to add these characteristics or modify the title of the table.
7. Results: Include tables 2 and 3 of the manuscript in this section and achieve a link with the text.
8. Discussion: The first paragraphs of the discussion seem more like results than actual discussion. I advise the authors to add a paragraph on the limitations and strengths of the study and another on the practical implications that can be drawn from the results.
9. Conclusion: A conclusion section should be added. There should be as many conclusions as objectives of the study.
Thank you very much. Yours sincerely
Reviewer 3 Report
Comments and Suggestions for Authors
Strengths: By studying the correlation between various pediatric assessments, we have proven the relevance between assessment methods.
1. Please check the spelling and grammar of the text.
2.. Please provide references for all sentences presented in the introduction.
3. Can you do sample size calculation?
4. Were they evaluated by raters blinded to the study?
5. Please fill out the reliability and validity of the evaluation tool.
6. Please add more clinical significance to the discussion of this article.
7. Is there no normality test?
8. Please write down the statistical method.
Comments on the Quality of English LanguagePlease check the spelling and grammar of the text.
